# The evidence supporting AHA guidelines on adult cardiopulmonary resuscitation (CPR)

**Emma Junedahl**[1]*, **Peter Lundgren**[1], **Erik Andersson**[1], **Vibha Gupta**[1], **Truls Råmunddal**[1], **Aidin Rawshani**[1], **Araz Rawshani**[1,2], **Gabriel Riva**[3], **Ida Arnetorp**[1], **Fredrik Hessulf**[2], **Johan Herlitz**[2], **Therese Djärv**[3]

1 Department of Molecular and Clinical Medicine, Institute of Medicine University of Gothenburg, Gothenburg, Sweden, 2 The Swedish Cardiopulmonary Resuscitation Registry, Gothenburg, Sweden, 3 Center for Resuscitation Science, Department of Science and Education KI SOS, Karolinska Institutet, Solna, Sweden

* ejunedahl@gmail.com

**Data Availability Statement:** We confirm that our submission contains all the raw data required to

## Abstract

### Background

Guidelines for the management of cardiac arrest play a crucial role in guiding clinical decisions and care. We examined the strength and quality of evidence underlying these recommendations in order to elucidate strengths and gaps in knowledge.

### Methods

Using the 2020 American Heart Association (AHA) Guidelines for Adult CPR, we subdivided all recommendations into advanced life support (ALS), basic life support (BLS), and recovery after cardiac arrest, as well as a more granular categorization by topic (i.e. the intervention or evaluation recommended). The Class of Recommendation (COR) and Level of Evidence (LOE) for each were reviewed. Additionally, we reviewed the 2023 guidelines to ensure the inclusion of the most recent updates.

### Results

We noted 254 recommendations, of which 181 were ALS, 69 were BLS, and 4 were recovery after resuscitation. In total, only 2 (1%) had the most robust evidence (LOE A), while 23% were at LOE B-NR (Non-Randomized), 15% at LOE B-R (Randomized), 50% at LOE C-LD (Limited Data), and 12% relied on expert opinion LOE C-EO (Expert Opinion). Despite the strength of ALS recommendations (Class 1, 2a, or 2b), none had LOE A. In BLS, no recommendations were supported by LOE A. For BLS, 7% of recommendations had LOE C (C-LD or C-EO). The evidence for specific BLS topics, such as airway management, was notably low. Among ALS topics, neurological prognostication had relatively stronger evidence.

### Conclusions

Only 26 out of the 81 COR 1 recommendations (32%) were supported by LOE A or B, indicating a strong discrepancy between the strength of recommendation and the underlying

replicate the results of our study, as well as the values used to build graphs. All relevant data are provided within the manuscript and its Supporting information files.

**Funding:** The author(s) received no specific funding for this work.

**Competing interests:** The authors have declared that no competing interests exist.

evidence in cardiac arrest guidelines. The findings underscore a pressing need for more rigorous research, particularly randomized trials.

## Introduction

Cardiac arrest impacts 400,000 Europeans and 350,000 Americans annually, of which 90% succumb [1]. The International Liaison Committee on Resuscitation (ILCOR) issues evidence-based treatment recommendations for all aspects of cardiopulmonary resuscitation. Every five years, ILCOR releases updated summaries of science and treatment recommendations on cardiopulmonary resuscitation (CPR) and emergency cardiovascular care to mirror the most recent scientific advances [2]. The European Resuscitation Council (ERC) and American Heart Association (AHA) use the ILCOR statements to develop clinical practice guidelines.

It's important to understand the context where AHA's guidelines for adult CPR are relevant and how they are utilized within emergency care. AHA, along with other organizations like the European Resuscitation Council (ERC), plays a crucial role by providing evidence-based guidelines to improve survival following cardiac arrest and other acute cardiovascular events. These guidelines are designed to offer healthcare professionals clear and scientifically grounded recommendations to enhance patient outcomes and standardize care globally.

Part 3 of AHA's guidelines for CPR and emergency cardiovascular care specifically covers guidelines for adult CPR and related emergency care. This section focuses on providing evidence-based recommendations for managing cardiac arrest and acute cardiovascular events in adult patients. The content of Part 3 encompasses various aspects of emergency care, including assessment, airway and respiratory support, chest compressions, defibrillator use, and post-ROSC care (Return of Spontaneous Circulation). These guidelines are of paramount importance in guiding healthcare professionals to deliver adequate and effective care during life-threatening situations such as cardiac arrest.

Recommendations of the American Heart Association are divided into different classes of recommendation (COR) and levels of evidence (LOE). The COR rating reflects the benefit-risk balance and corresponds to the strength of the recommendation. Class 1 indicates that an assessment or intervention is useful, effective, and should be used. Class 2 recommendations are weaker and have a lower degree of benefit compared to class 1. For Class 2a, the benefit is estimated to marginally exceed the risk. Class 2b recommendations should be used selectively and individually, as the benefit is questionable. Class 3 implies that the potential benefit is as great as the risk.

Level A (LOE-A) involves high-quality evidence from multiple randomized controlled trials (RCT), meta-analyses of high-quality RCTs, and one or more RCTs supported by high-quality registry studies. Level B-Randomized (LOE B-R) is based on moderately qualitative evidence from one or more RCTs or meta-analyses of moderate-qualitative RCTs. Level B-Non Randomized (LOE B-NR) entails moderate qualitative evidence from one or more well-designed and well-performed non-randomized studies. Level C-Limited Data (LOE C-LD) is based on limited data and includes either randomized or non-randomized observational or registry studies with limitations in design or execution. Level C-Expert Opinion (LOE C-EO) is based on expert opinion [3].

A review published in 2009 of the ACC/AHA guidelines between 1984–2008 showed that 11% of the recommendations were classified as LOE A [4]. A review of the recommendations issued between 2008–2018, showed that the corresponding proportion was 9% [5]. We

conducted a renewed survey of the latest guidelines, published in 2020 to map the development of the evidence as well as variations in different aspects of CPR in adults.

## Methods

Data were gathered from the American Heart Association's (AHA) Guidelines for CPR and Emergency Cardiovascular Care 2020, Part 3 Adult CPR [3]. All individual recommendations issued in 2020 were inclusively collected and assigned to specific categories corresponding to the topic area. The categories encompassed the following domains: Accidental hypothermia, Adjuncts to CPR, Airway management, Amiodoarone or lidocaine, Anaphylaxis or asthma, Antiarrhythmic agents excluding amiodarone and lidocaine, BLS recognition and initiation of CPR, Cardiac arrest due to overdose and toxicity, Cardiac arrest in cardiac surgery, Cardiac arrest in special circumstances, Chest compressions, Coronary angiography, CPR feedback, monitoring, checks, Defibrillation, Drowning, ECMO (extracorporeal membrane oxygenation), Electrolyte abnormalities in cardiac arrest, Glycemic control post-ROSC, Hemodynamic managementpost-ROSC, Initiation of CPR, Management of bradycardia, Management of supra ventricular arrythmias, Neuroprognostication, Other, Other antiarrythmic interventions, Other pharmacological agents, Positioning during CPR, Post-ROSC diagnostics, Pregnancy, Prophylactic antibiotics post-ROSC, Pulmonary embolism, Recovery and survivorship, Seizure management, Steroids, Termination of resuscitation, Targeted temperature management (TTM), Vascular access, Vasopressors, Ventilation, Ventilation post-ROSC.

The recommendations were further categorized into subgroups delineating advanced life support (ALS), basic life support (BLS), and recovery. Within each category, the classification of recommendations (COR) and level of evidence (LOE) were scrutinized to explore the strength of the recommendation and the underlying scientific support. For each recommendation, its classification according to COR and LOE, along with its associated category and subcategory, were documented. This information was utilized to assess the overall strength and scientific support of the adult CPR guidelines as per the AHA's 2020 guidelines. This approach was conducted in accordance with the method of Fanaroff et al [5], which facilitated straightforward abstraction since each recommendation has a designated COR and LOE, thus only requiring reporting.

The recommendations collected were analyzed using the R programming language. R facilitated the quantitative assessment of the recommendations' levels of evidence (LOE) and their corresponding classifications of recommendations (COR). Through this analysis, we were able to comprehensively quantify and visualize the distribution of LOE and COR across the primary categories of advanced life support (ALS), basic life support (BLS), and recovery. By utilizing R's statistical and graphical capabilities, we ensured precise and reproducible quantification, which enabled the generation visual representations. The data visualization process involved creating bar plots and cross-tabulations to depict the proportion of recommendations across different COR and LOE categories, providing an overview of the evidence base supporting each recommendation. The data was processed both as an aggregate and stratified by topic.

To ensure the presentation of current data, we reviewed the updated guidelines from 2023 published by Perman et al, as this was the first formal update since the publication of the 2020 guidelines. According to Perman et al, this focused update addressed new evidence and made revisions to certain recommendations [6]. Specifically, we examined changes in the class of recommendation (COR) and level of evidence (LOE) for all the recommendations we studied in the 2020 edition. The changes identified from the 2023 update are presented in the Results

section and have not been included in the data visualization to maintain the integrity of the 2020 dataset.

## Results

Among 254 recommendations, there were 181 recommendations for ALS, 69 recommendations for BLS, and 4 recommendations for Recovery after cardiac arrest.

Table 1 **shows that** overall recommendations had the following distribution: 0.8% (2) had LOE A, 15% (37) had LOE B-R, 23% (58) had LOE B-NR, 50% (126) had LOE C-LD, and 12% (31) had LOE C-EO.

ALS recommendations had the following distribution: 1% (2) had LOE A, 17% (30) had LOE B-R, 25% (46) had LOE B-NR, 46% (84) had LOE C-LD, and 10% (19) had LOE C-EO.

BLS recommendations had the following distribution: 0% (0) had LOE A, 10% (7) had LOE B-R, 16% (11) had LOE B-NR, 57% (39) had LOE C-LD, and 17% (12) had LOE C-EO.

There were 81 class 1 recommendations (32% of total). Among those, 0% (0/81) had LOE A, 9% (7/81) had LOE B-R, 24% (19/81) had LOE B-NR, 48% (39/81) had LOE C-LD, 20% (16/81) had LOE C-EO.

There were 58 Class 2a recommendations (23% of the total). Among those, 0% (0/58) had LOE A, 17% (10/58) had LOE B-R, 24% (14/58) had LOE B-NR, 52% (30/58) had LOE C-LD, and 7% (4/58) had LOE C-EO.

**Table 1. Level of evidence (LOE) and class of recommendation (COR) for all recommendations, ALS recommendations and BLS recommendations.**

| All recommendations | Level of Evidence | | | | | Total |
|---|---|---|---|---|---|---|
| | **A** | **B-R** | **B-NR** | **C-LD** | **C-EO** | |
| COR | | | | | | |
| 1 | 0 (0%) | 7 (2.8%) | 19 (7.5%) | 39 (15%) | 16 (6.3%) | 81 (32%) |
| 2a | 0 (0%) | 10 (3.9%) | 14 (5.5%) | 30 (12%) | 4 (1.6%) | 58 (23%) |
| 2b | 0 (0%) | 13 (5.1%) | 19 (7.5%) | 48 (19%) | 9 (3.5%) | 89 (35%) |
| 3: No benefit | 2 (0.8%) | 6 (2.4%) | 3 (1.2%) | 4 (1.6%) | 0 (0%) | 15 (5.9%) |
| 3: Harm | 0 (0%) | 1 (0.4%) | 3 (1.2%) | 5 (2.0%) | 2 (0.8%) | 11 (4.3%) |
| *Total* | *2 (0.8%)* | *37 (15%)* | *58 (23%)* | *126 (50%)* | *31 (12%)* | *254 (100%)* |
| **ALS** | Level of Evidence | | | | | Total |
| | **A** | **B-R** | **B-NR** | **C-LD** | **C-EO** | |
| COR | | | | | | |
| 1 | 0 (0%) | 7 (3.9%) | 13 (7.2%) | 26 (14%) | 11 (6.1%) | 57 (31%) |
| 2a | 0 (0%) | 6 (3.3%) | 10 (5.5%) | 19 (10%) | 2 (1.1%) | 37 (20%) |
| 2b | 0 (0%) | 11 (6.1%) | 17 (9.4%) | 34 (19%) | 4 (2.2%) | 66 (36%) |
| 3: No benefit | 2 (1.1%) | 5 (2.8%) | 3 (1.7%) | 2 (1.1%) | 0 (0%) | 12 (6.6%) |
| 3: Harm | 0 (0%) | 1 (0.6%) | 3 (1.7%) | 3 (1.7%) | 2 (1.1%) | 9 (5.0%) |
| *Total* | *2 (1.1%)* | *30 (17%)* | *46 (25%)* | *84 (46%)* | *19 (10%)* | *181 (100%)* |
| **BLS** | Level of Evidence | | | | | Total |
| | **A** | **B-R** | **B-NR** | **C-LD** | **C-EO** | |
| COR | | | | | | |
| 1 | 0 (0%) | 0 (0%) | 5 (7.2%) | 11 (16%) | 5 (7.2%) | 21 (30%) |
| 2a | 0 (0%) | 4 (5.8%) | 4 (5.8%) | 11 (16%) | 2 (2.9%) | 21 (30%) |
| 2b | 0 (0%) | 2 (2.9%) | 2 (2.9%) | 13 (19%) | 5 (7.2%) | 22 (32%) |
| 3: No benefit | 0 (0%) | 1 (1.4%) | 0 (0%) | 2 (2.9%) | 0 (0%) | 3 (4.3%) |
| 3: Harm | 0 (0%) | 0 (0%) | 0 (0%) | 2 (2.9%) | 0 (0%) | 2 (2.9%) |
| *Total* | *0 (0%)* | *7 (10%)* | *11 (16%)* | *39 (57%)* | *12 (17%)* | *69 (100%)* |

There were 89 Class 2b recommendations (35% of the total). Among those, 0% (0/89) had LOE A, 15% (13/89) had LOE B-R, 21% (19/89) had LOE B-NR, 54% (48/89) had LOE C-LD, and 10% (9/89) had LOE C-EO.

There were 15 Class 3: No benefit recommendations (6% of the total). Among those, 13% (2/15) had LOE A, 40% (6/15) had LOE B-R, 20% (3/15) had LOE B-NR, 27% (4/15) had LOE C-LD, and 0% (0/15) had LOE C-EO.

There were 11 Class 3: Harm recommendations (4% of the total). Among those, 0% (0/11) had LOE A, 9% (1/11) had LOE B-R, 27% (3/11) had LOE B-NR, 18% (2/11) had LOE C-LD, and 18% (2/11) had LOE C-EO.

Fig 1 illustrates the distribution of class of recommendation and level of evidence for ALS, BLS, and recovery after cardiopulmonary resuscitation. We note that very few recommendations suggest harm or absence of benefit, whereas the majority suggest clear benefit (COR 1) or probable benefit (COR 2a). Yet, a negligible portion (1%) of these recommendations were

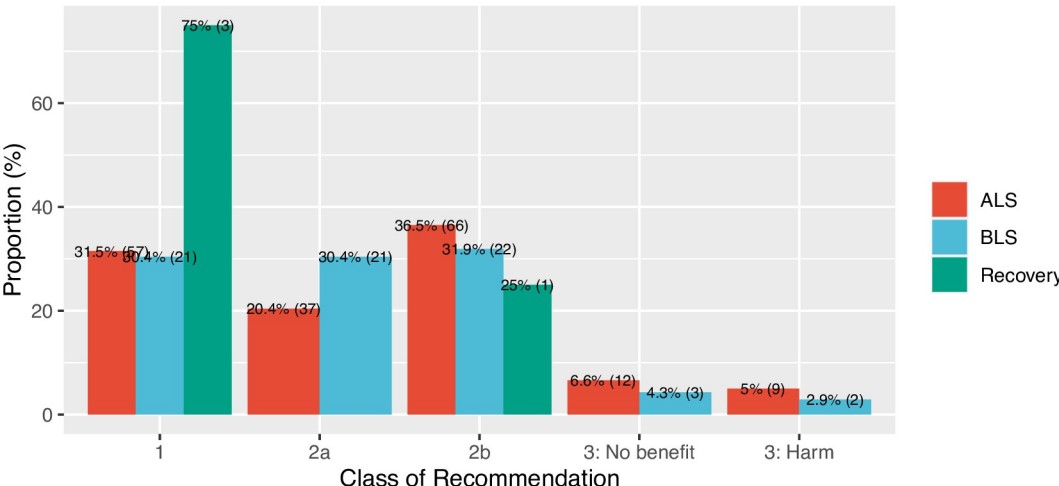

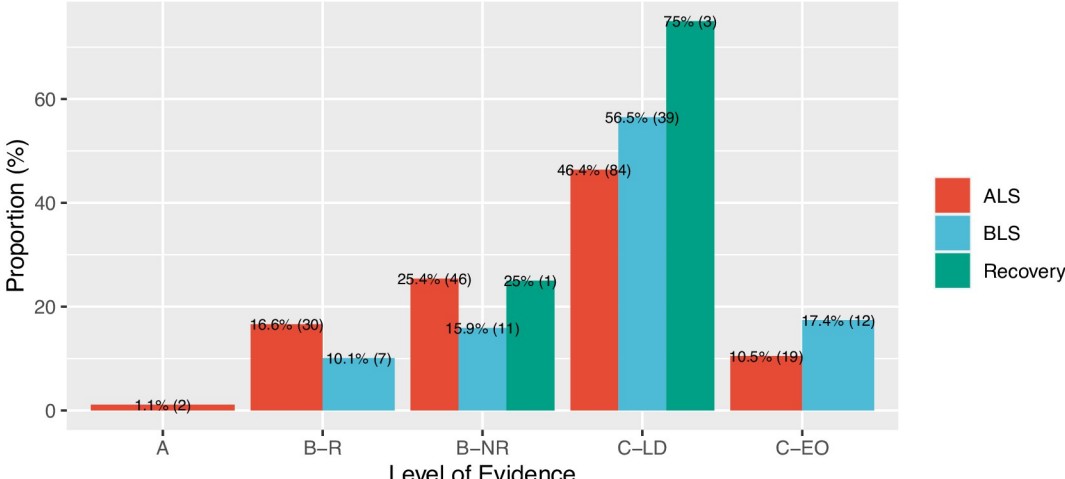

**Fig 1. Distribution of COR and LOE across ALS, BLS and recovery after CPR.** The figure illustrates the distribution of recommendations by class (1, 2a, 2b, 3: No benefit, and 3: Harm) and level of evidence (A, B-R, B-NR, C-LD, and C-EO) for advanced life support (ALS), basic life support (BLS), and recovery post-cardiopulmonary resuscitation.

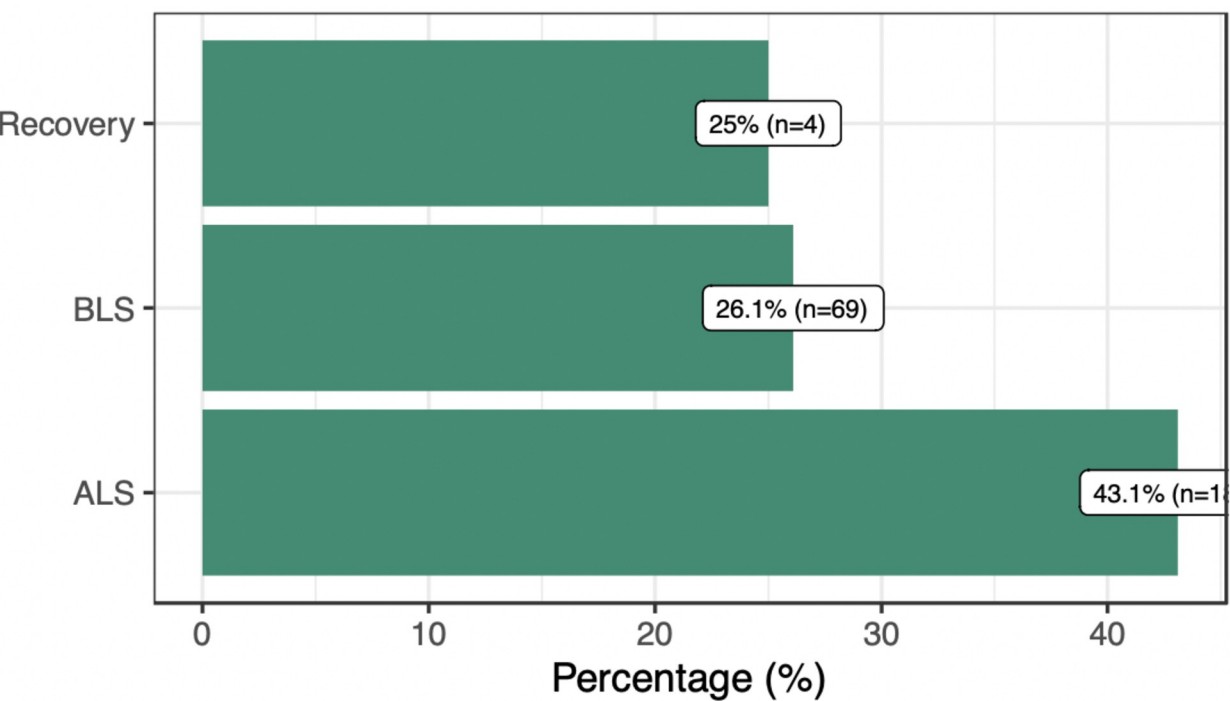

**Fig 2. Proportion of recommendations that have LOE A or B.** The figure shows the percentage of recommendations with level of evidence (LOE) A or B across advanced life support (ALS), basic life support (BLS) and recovery post-cardiopulmonary resuscitation.

based on firm evidence (LOE A). Overall, 26% of BLS recommendations had LOE A or B, as compared with 43% for ALS.

Fig 2 illustrates the distribution of recommendations for advanced life support (ALS), basic life support (BLS), and recovery after cardiac arrest, based on the class of recommendation and level of evidence. We observe that a small percentage of the recommendations, specifically 25% for recovery, 26% for BLS, and 43% for ALS, are based on high-quality evidence (Level A or B).

Fig 3 displays the topic-specific level of evidence and class of recommendation. For BLS, all recommendations for airway management, positioning, and ventilation were based on low evidence (LOE C). Among BLS recommendations for defibrillation, 5 out of 16 recommendations had LOE B (B-NR or B-R). For ALS, neuroprognostication stood out as the most evidence-based topic, although all recommendations with LOE B were of type B-NR (i.e., the lower grade of LOE B).

Fig 4 shows the proportion of all recommendations, by topic, which had at least LOE B (either B-R or B-NR). Recommendations for management post ROSC had a clearly higher evidence grade.

During the review of the 2023 updates to the recommendations, we identified the following changes regarding class of recommendation (COR) and level of evidence (LOE):

For vasopressors, the 2020 guidelines strongly recommended the use of epinephrine during cardiac arrest (COR 1), while vasopressin was not specifically mentioned. In the 2023 guidelines, the recommendation for epinephrine remains (COR 1), but vasopressin, alone or in combination with methylprednisolone, is now considered as an adjunct (COR 2b).

Regarding antiarrhythmic drugs, the 2020 guidelines suggested that amiodarone or lidocaine could be considered for ventricular fibrillation/pulseless ventricular tachycardia that is

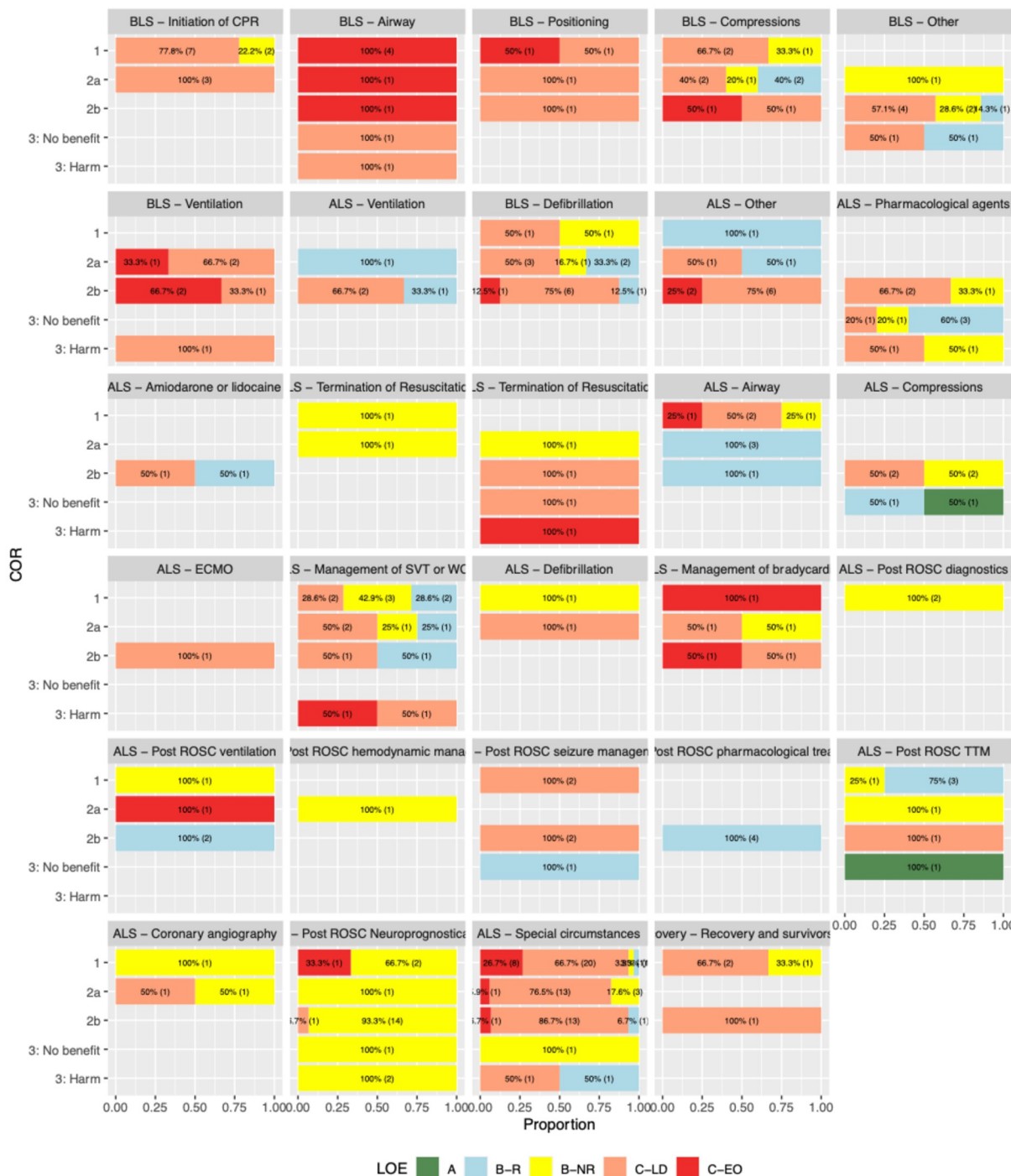

**Fig 3. Distribution of LOE and COR across various topics.** The figure illustrates the distribution of levels of evidence (LOE) and classes of recommendations (COR) across multiple topics. Each panel represents a different topic, with the proportions of recommendations categorized by LOE (A, B-R, B-NR. C-LD, C-EO) and COR (1, 2a, 2b, 3: No benefit, 3: Harm) displayed.

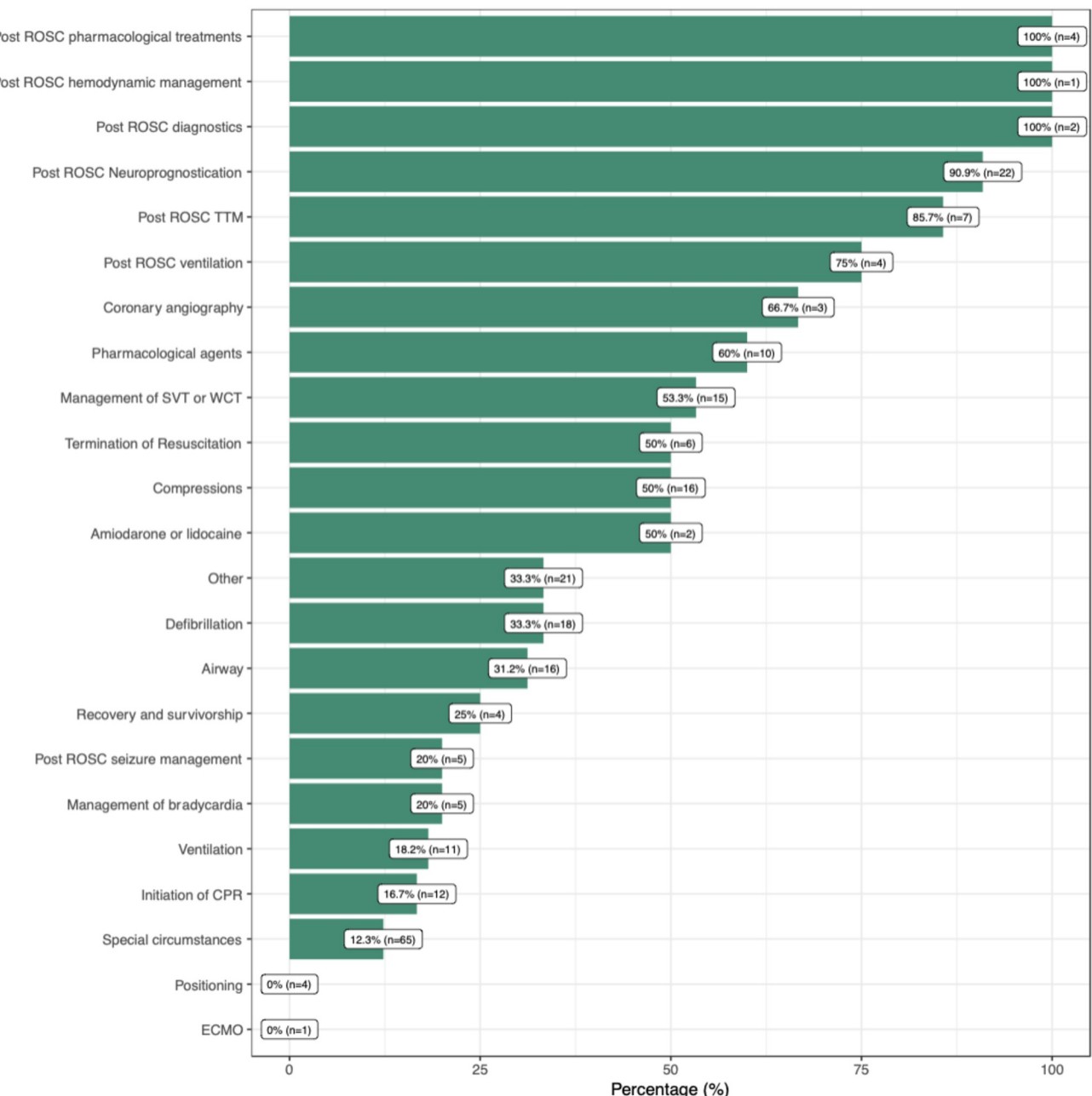

**Fig 4. Proportion of recommendations with LOE A or B across various topics.** The figure illustrates the percentage of recommendations within each topic that are supported by level of evidence (LOE) A or B. The topics are arranged in descending order based on the proportion of high-evidence recommendations.

unresponsive to defibrillation (COR 2b). The 2023 update maintains this recommendation (COR 2b), but adds that the routine administration of calcium, sodium bicarbonate, and magnesium during cardiac arrest is not recommended.

For extracorporeal cardiopulmonary resuscitation (ECPR), the 2020 guidelines considered its use in select cases (COR 2b). The 2023 guidelines now deem ECPR reasonable for patients

with cardiac arrest refractory to standard ACLS when equipment and trained personnel are available (COR 2a).

The recommendation for coronary angiography in the 2020 guidelines was for patients with suspected cardiac cause and ST-segment elevation (COR 1). The 2023 guidelines continue to recommend coronary angiography for patients with ST-segment elevation (COR 1) and now also consider it reasonable for high-risk patients without ST-segment elevation (COR 2a).

Finally, the 2020 guidelines recommended targeted temperature management (TTM) between 32˚C and 36˚C for post-cardiac arrest patients (COR 1). The 2023 update has revised this term to "temperature control" and expanded the recommended range to 32˚C to 37.5˚C (COR 1).

## Discussion

The evidence base for current guidelines on the management of cardiac arrest reveals a strong discrepancy between the strength class of recommendation and the underlying level of evidence. While a majority of recommendations suggest a clear or probable benefit (Class 1 and Class 2a), very few are backed by robust evidence. Such a mismatch underscores the challenges in producing high-quality evidence in the hyperacute, real-time scenarios of cardiac arrest.

Among ALS recommendations, we note the absence of high-level evidence (LOE A) despite a high prevalence of strong recommendations (Class 1, 2a, or 2b). Over half of the ALS recommendations were based on limited data or expert opinions, which, while valuable and perhaps clinically sound, do not provide the rigorous evidence essential for such critical care settings. We recognize that lack of reliable evidence may have little clinical significance for some interventions (e.g. the recommendation to record a 12-lead ECG upon ROSC), but this study demonstrates an array of critical interventions and tasks that lack a sound evidence-base as demonstrated in Fig 3.

This simple study elucidates the significant gaps in our current understanding of cardiac arrest management, underscoring an urgent need for more studies focusing on these critical knowledge gaps. We emphasize that researchers should channel their efforts and resources towards more interventional studies, preferably in national and international collaborations to pool resources, expertise and power. We also recognize that while observational studies have their merits, there is a pressing need for more randomized trials and intervention studies.

With regards to BLS, perhaps the most widespread treatment ever introduced, there is an even more pronounced evidence deficit. With no recommendations supported by LOE A and a significant proportion relying on the weakest evidence types (LOE C-LD and LOE C-EO). We view this as a clear imperative to invest in more robust research within this domain, recognizing that there are many groups worldwide conducting excellent research in this field.

When examining the factors contributing to the low level of evidence for Class I and II recommendations across various categories, several potential causes emerge. A central aspect is the presence of clinical biases that can influence the formulation and level of evidence of recommendations [7]. These biases can lead to overvaluation or undervaluation of certain evidence, affecting the reliability and level of evidence of recommendations. Secondly, system-related factors can also play a significant role. Political or economic interests, for example, can influence decision-making regarding guidelines and recommendations, prioritizing certain treatments despite the lack of sufficient scientific support [8]. This can result in a bias in the available evidence, thereby contributing to the low level of evidence for certain recommendations.

Our study is in line with research in other cardiovascular conditions, as demonstrated by Tricoci et al [4] and Fanaroff et al, who showed that only a minority of cardiovascular

guidelines were based on LOE A [5]. The Evolution of the ACC/AHA Clinical Guidelines study indicated a decrease in LOE-C recommendations between 2008 and 2018, implying a strengthening of the guidelines by removing lower-quality recommendations. However, there has not been a corresponding increase in higher levels of evidence observed [9].

We also note relatively few Class 2b and Class 3 recommendations (lack of effect or potential harm) within the guidelines, which we believe is a reflection of the well-documented issue in scientific research: the under-reporting of negative or null results. Studies that fail to demonstrate significant beneficial outcomes or show potential adverse effects are often less likely to be submitted, accepted, or published compared to those with positive outcomes. We find no reason to believe the cardiac arrest field would differ from other fields regarding this, and it is presumably an important explanation for the distortion in the distribution demonstrated here. This gap not only affects the reliability of guidelines but also deprives clinicians and researchers of vital information that could guide clinical decision-making and future research.

Upon reviewing the 2023 updates to the adult advanced cardiovascular life support, we identified that few changes have been made to the Class of Recommendation (COR) and Level of Evidence (LOE) compared to the 2020 guidelines. The updates primarily include the expanded use of vasopressin, additional guidance on the use of antiarrhythmic drugs, strengthened recommendations for extracorporeal cardiopulmonary resuscitation (ECPR), and a broader range for temperature control. These changes, although noteworthy, have minimal impact on the overall conclusions presented in this study. The fundamental principles and core recommendations remain consistent, reinforcing the robustness and applicability of our findings based on the 2020 guidelines.

In conclusion, the prevailing evidence base accentuates a pressing need for enhanced research efforts, with more focus on interventional trials. Both ALS and BLS recommendations, crucial in the hyperacute context of cardiac arrest, warrant more rigorous and high-quality evidence from randomized trials.

From a clinical perspective, our study underscores the importance of critical evaluation of current guidelines and their evidence base. Clinicians should be aware of the potential limitations and discrepancies between recommendation strength and underlying evidence when making patient care decisions. Future research should aim to fill these evidence gaps to ensure that clinical practice is based on the most robust and up-to-date evidence available.

Furthermore, our study has implications for future research directions. Researchers should prioritize the generation of high-quality evidence through well-designed randomized controlled trials and intervention studies. Collaborative efforts, both nationally and internationally, are essential to effectively address the complexities of cardiac arrest management and improve patient outcomes.

## Limitations

We must consider that our categorization of recommendations into specific topics, such as ALS, BLS, and recovery, may have oversimplified the complexity of the guidelines. This detailed breakdown might not have fully captured all relevant nuances and variations within each topic, potentially resulting in a loss of detail in our analysis.

Furthermore, although we reviewed each recommendation's classification (COR) and level of evidence (LOE), our study was limited to evaluating the clinical guidelines in terms of their scientific foundation and recommended practice. We could not directly assess the implementation of these guidelines or their effectiveness in improving patient outcomes or the quality of care.

Additionally, our analysis did not include expert consensus documents, which can offer valuable insights, particularly in areas lacking robust empirical data. This omission should be considered a limitation of our study. Including these documents in future reviews will enhance the comprehensiveness and applicability of our findings.

Finally, we have not compared the AHA guidelines to recommendations from other organizations, such as the European Resuscitation Council (ERC). Such a comparison could provide additional insights and contexts for our understanding of best practices for CPR and emergency cardiovascular care.

## Supporting information

**S1 Appendix. Categorization of AHA CPR guidelines 2020.** This appendix includes the recommendations from the AHA 2020 guidelines, detailing the assignment of each recommendation to specific categories based on the relevant topic area.
(CSV)

**S1 File.**
(PDF)

## Author Contributions

**Conceptualization:** Araz Rawshani.

**Formal analysis:** Emma Junedahl.

**Investigation:** Emma Junedahl.

**Methodology:** Emma Junedahl, Araz Rawshani.

**Supervision:** Araz Rawshani.

**Visualization:** Emma Junedahl.

**Writing – original draft:** Emma Junedahl.

**Writing – review & editing:** Peter Lundgren, Erik Andersson, Vibha Gupta, Truls Råmunddal, Aidin Rawshani, Araz Rawshani, Gabriel Riva, Ida Arnetorp, Fredrik Hessulf, Johan Herlitz, Therese Djärv.

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
