## [Decision Letter · Decision Letter 0]

20 Mar 2024

PONE-D-24-02668The evidence supporting AHA guidelines on adult cardiopulmonary resuscitation (CPR)PLOS ONE

Dear Dr. Junedahl,

Thank you for submitting your manuscript to PLOS ONE. After careful consideration, we feel that it has merit but does not fully meet PLOS ONE’s publication criteria as it currently stands. Therefore, we invite you to submit a revised version of the manuscript that addresses the points raised during the review process.

We look forward to receiving your revised manuscript.

Kind regards,

Ankur Shah

Academic Editor

PLOS ONE

Journal Requirements:

5. Please amend the manuscript submission data (via Edit Submission) to include authors Dr. Peter Lundgren, Dr. Erik Andersson, Dr. Vibha Gupta1, Dr. Truls Råmunddal, Dr. Aidin Rawshani, Dr. Gabriel Riva, Dr.Ida Arnetorp, Dr. Fredrik Hessulf, Dr. Johan Herlitz and Dr. Therese Djärv.

6. One of the noted authors is a group or consortium. In addition to naming the author group, please list the individual authors and affiliations within this group in the acknowledgments section of your manuscript. Please also indicate clearly a lead author for this group along with a contact email address.

7. Please include your tables as part of your main manuscript and remove the individual files. Please note that supplementary tables (should remain/be uploaded) as separate "supporting information" files.

Additional Editor Comments:

The authors submission evaluates the quality of evidence underlying the guidelines of resuscitation, they do not need to address the field of guideline analysis and can focus their revisions to the 2nd and 3rd reviewers comments.

Reviewers' comments:

Reviewer's Responses to Questions

**Comments to the Author**

1. Is the manuscript technically sound, and do the data support the conclusions?

Reviewer #1: No

Reviewer #2: Partly

Reviewer #3: Yes

2. Has the statistical analysis been performed appropriately and rigorously? 

Reviewer #1: I Don't Know

Reviewer #2: Yes

Reviewer #3: N/A

3. Have the authors made all data underlying the findings in their manuscript fully available?

Reviewer #1: Yes

Reviewer #2: Yes

Reviewer #3: Yes

4. Is the manuscript presented in an intelligible fashion and written in standard English?

Reviewer #1: No

Reviewer #2: Yes

Reviewer #3: Yes

5. Review Comments to the Author

Reviewer #1: The methodology of the study does not have a clear definition point and outcome.

In CPR studies, if the causes of arrest are classified within themselves and a parallelism between treatment protocols is shown, better quality results can be obtained. The diagnoses after ROSC and the quality of ALS and BLS applied before reveal the quality of these guidelines. However, I believe that comparisons and ratios of such studies would be a better literature study.

The discussion cannot provide a complex and holistic result due to the weakness of the method.

Reviewer #2: I commend the authors for their thorough analysis of the evidence backing the AHA guidelines on adult CPR. Authors emphasize that a little 0.8% of adult CPR recommendations were supported by Level of Evidence A, but almost 62% of recommendations were based on Level of Evidence C. The study offers insights into the discrepancy between the recommendation grade and the quality of supporting evidence from studies and experts.

I have the following concerns regarding the manuscript:

1. Kindly explain the context in which these guidelines are relevant and specify the content covered in "part 3" of the AHA guidelines for CPR and emergency cardiovascular care.

2. Please explain the process of reviewing and selecting guidelines. Can the authors provide the total number of guidelines being investigated? Did the writers assess all the available guidelines or focus on a specific set? Were predetermined criteria utilized to select the guidelines?

3. The paper mainly talks about the 2020 AHA guidelines for CPR along with the additional changes in guidelines from 2021 and 2022. Can the authors specify which guidelines were altered and the extent of these alterations? How was a modification in the recommendation class or evidence level integrated into the analysis? Were the authors able to compare the revised outcome with the initial (2020) recommendation/evidence to assess the effect of the modification?

4. Please include the relevant information on the "Recovery after CPR" guidelines in the paper.

5. Please incorporate the pertinent details from Figure 1b into the manuscript text.

6. No mention or comparison is made with previous years CPR recommendations or guidelines from other societies like the European Resuscitation Council. The authors have not taken into the account the evolution of evidence quality in the current recommendations over time. Please consider adding this discussion point or addressing them in the Limitations section.

7. Kindly outline the Limitations of the current study

8. Kindly provide the clinical and future research implications of this study.

9. Please elaborate on the factors contributing to the low level of evidence for Class I and II recommendations across various categories. Can current clinical biases influence these recommendations? Can the authors cite relevant literature to support this discussion with respect to their current work?

10. Some MINOR corrections:

A) Please provide the full form of "ROSC"

B) Ensure that percentages are presented consistently in the text, either as whole numbers (e.g., 26%) or with one decimal point (e.g., 26.1%).

C) Consider rewriting the following in Introduction to avoid confusion:

> Explanation of Class 2 recommendations (Class 2 recommendations are weaker "and have a lower degree of benefit, compared to risk"). Does the class 2 recommendation have a lower degree of benefit as compared to Class I or when compared to the risk of the intervention?

> Explanation of Class 3 recommendations (Class 3 implies that the potential benefit is as great as the risk, which

suggests the "absence of evidence" or risk of harm). The authors talk about the absence of evidence of a benefit from the recommendation?

D) Please highlight the "Figure" and "Table" within the text.

E) Please ensure that citations are in the correct reference style and in consistent format (e.g., in the line "Our study is in line with research in other cardiovascular conditions, as demonstrated by Tricoci et al,4 Fanaroff et al, who showed that only a minority of cardiovascular guidelines were based on LOE A.5," either have the citations after "et al" or at the end of the sentence).

F) A suggestion: Providing "Figure 2" or "Figure 3" citation at the end of the sentence "but this study demonstrates an array of critical interventions and tasks that lack a sound evidence-base." in Discussion can provide context to the reader.

Thank you.

Reviewer #3: The authors should be congratulated for their thoughtful submission.

It is indeed an insight into the literature to support the guideline development process, and the overall transparency of that process.

Some minor suggestions are listed below.

Abstract: re “Overall, 26.1% of BLS recommendations had LOE A or B, versus 43.1% for ALS recommendations.” Suggest delete this as potentially confusing: just relates to LOE B as already stated that none for ALS or BLS were supported by LOE A.

Abstract: re “There is a strong discrepancy between the strength of recommendation and the underlying evidence in cardiac arrest guidelines.” This is not described in the rest of the abstract: this would require listing of strength versus level of evidence. Please amend accordingly. Could include statement in abstract such as “Of the 81 COR 1 recommendations only 26 (32%) were supported by LOE A or B.”

Introduction: re “The International Liaison Committee on Resuscitation (ILCOR) issues

evidence-based treatment guidelines”. ILCOR publishes Treatment Recommendations not Guidelines: please amend this.

Introduction: re “ILCOR releases updated guidelines”. ILCOR releases “summaries of science and treatment recommendations”: please amend this.

Introduction: re “Recommendations are divided . . . ”. This relates to the AHA COR and LOE, so should be rewritten as “Recommendations of the American Heart Association are divided . . .”.

Methods: Anaphylaxis is misspelt.

Methods: “Hemodynamic management, post-ROSC” should read “Hemodynamic

management post-ROSC”

Methods: “Management of supra ventricular arrythmias” should read “Management of

supra-ventricular arrythmias”

Methods: “Post ROSC diagnostics” should read “Post-ROSC diagnostics”

Results: the use of denominators can be confusing. Re “There were in total 32% (81) Class 1 recommendations. Among those, 0% (0) had LOE A, 2.8% (7) had LOE B-R, . . .” suggest this be reworded to “There were 81 Class 1 recommendations (32% of total). Among those, 0% (0/81) had LOE A, 8.6% (7/81) had LOE B-R, . . .”. This makes the proportions more interpretable and is what the readers need to see to support your argument.

Results: suggest the same approach as above be used for the listing of summary data/outcomes for Class 2a, Class 2b, and both Class C recommendations that follow the Class 1 paragraph.

Discussion: re “Such a mismatch underscores the challenges and gaps in producing high-quality evidence”. Suggest delete “and gaps” as this does not read well, and is implied by the previous sentence.

Discussion: re “this is suggested by the difference in the proportion of recommendations with LOE A or B for the pre- and post-ROSC phase.” Suggest this can be deleted, as this is not necessary.

Discussion: re “absence of reliable evidence”. Suggest this read “absence of high-level evidence”.

Discussion: re “Tricoci et al,4 Fanaroff et al,”. Suggest thss read “Tricoci et al4 and Fanaroff et al,”.

Discussion: re “Althought, the Evolution of the ACC/AHA Clinical Guidelines study

found that between 2008 and 2018, the ACC/AHA guidelines saw a decrease in LOE-C

recommendations, suggesting that the guidelines were gradually strengthened in that mening that recommendations based on lower quality (LOE-C) were removed. However, no corresponding increase in higher level of evidence has yet to been seen.” This sentence has spelling mistakes and does not flow. Suggest delete.

6. PLOS authors have the option to publish the peer review history of their article (what does this mean?). If published, this will include your full peer review and any attached files.

Reviewer #1: No

Reviewer #2: No

Reviewer #3: **Yes: **Peter Morley

---

## [Author Response · Author response to Decision Letter 0]

22 May 2024

Reviewer #1: 

Reviewer #1 raised concerns regarding the clarity of our methodology and the necessity for a more comprehensive analysis of CPR guidelines. In response, we have revised the methodology section to provide a clearer explanation of the guidelines' collection and categorization process. Additionally, to enhance clarity, we have included a separate document containing all guidelines in their entirety, illustrating that the only processing of the guidelines performed was their categorization into ALS/BLS/Recovery, along with the assignment of each guideline to a specific topic for streamlined data presentation. 

Due to the absence of updates to guidelines post-2020, we were unable to incorporate these subsequent revisions into our analysis. Furthermore, we have enhanced the discussion section to offer a more nuanced interpretation of our findings and their implications for future research. 

Reviewer #2: 

Thoroughness of Analysis: We have addressed the reviewer's request for additional context regarding the relevance of the guidelines and a more detailed explanation of the guideline selection process. In the revised manuscript, we have included a clear description of the topics covered by the guidelines and the rationale behind our categorization process. We have also provided information on how the guideline recommendations were reviewed and assessed for strength and evidence level. 

Clarity and Completeness: We have incorporated the reviewer's suggestions for improving the clarity and completeness of our manuscript. Specifically, we have included explanations of the context in which the guidelines are relevant, details on the guideline review process, and information on any modifications to the guidelines over time. Additionally, we have addressed each of the minor corrections pointed out by the reviewer. 

Reviewer #3: 

Insightfulness of Submission: We appreciate the reviewer's positive feedback on the insightfulness of our submission. We have carefully considered the reviewer's suggestions for minor revisions and have incorporated them into the revised manuscript. 

Additional Suggestions: We have addressed the reviewer's additional suggestions, including clarifications in the abstract, corrections to spelling and formatting errors, and enhancements to the discussion section. These changes have been implemented to improve the overall quality and readability of the manuscript.

---

## [Decision Letter · Decision Letter 1]

7 Jun 2024

PONE-D-24-02668R1The evidence supporting AHA guidelines on adult cardiopulmonary resuscitation (CPR)PLOS ONE

Dear Dr. Junedahl,

Thank you for submitting your manuscript to PLOS ONE. After careful consideration, we feel that it has merit but does not fully meet PLOS ONE’s publication criteria as it currently stands. Therefore, we invite you to submit a revised version of the manuscript that addresses the points raised during the review process.

We look forward to receiving your revised manuscript.

Kind regards,

Ankur Shah

Academic Editor

PLOS ONE

Journal Requirements:

Additional Editor Comments:

The authors are requested to revise in accordance with the following reviewer request

Kindly provide a more comprehensive explanation of the Data extraction and abstraction process and the statistical analysis in Methodology. Kindly refer to the following articles for reference:

Gonzalez-Del-Hoyo M, Mas-Llado C, Blaya-Peña L, et al. Type of evidence supporting ACC/AHA and ESC clinical practice guidelines for acute coronary syndrome. Clin Res Cardiol. 2024;113(4):546-560. doi:10.1007/s00392-023-02262-9

Fanaroff AC, Califf RM, Windecker S, Smith SC Jr, Lopes RD. Levels of Evidence Supporting American College of Cardiology/American Heart Association and European Society of Cardiology Guidelines, 2008-2018. JAMA.

Reviewers' comments:

Reviewer's Responses to Questions

**Comments to the Author**

1. If the authors have adequately addressed your comments raised in a previous round of review and you feel that this manuscript is now acceptable for publication, you may indicate that here to bypass the “Comments to the Author” section, enter your conflict of interest statement in the “Confidential to Editor” section, and submit your "Accept" recommendation.

Reviewer #1: (No Response)

Reviewer #2: All comments have been addressed

Reviewer #3: All comments have been addressed

2. Is the manuscript technically sound, and do the data support the conclusions?

Reviewer #1: No

Reviewer #2: Partly

Reviewer #3: Yes

3. Has the statistical analysis been performed appropriately and rigorously? 

Reviewer #1: I Don't Know

Reviewer #2: Yes

Reviewer #3: N/A

4. Have the authors made all data underlying the findings in their manuscript fully available?

Reviewer #1: Yes

Reviewer #2: Yes

Reviewer #3: Yes

5. Is the manuscript presented in an intelligible fashion and written in standard English?

Reviewer #1: No

Reviewer #2: Yes

Reviewer #3: Yes

6. Review Comments to the Author

**Reviewer #1:** (No Response)

**Reviewer #2:** Dear Authors, thank you for providing a thorough revision of the manuscript. Although most of my concerns were answered, I have a few further points too be considered:

1. Are focused updates included in the revised manuscript? Consider the recently published 2023 updates, along with the previously considered 2021 and 2022 updates, of the guidelines to seek any modification in the guidelines from 2020. This would help the article to be up-to-date with the most recent changes and current scenarios and better guide readers. The article can be found as follows:

Perman SM, Elmer J, Maciel CB, et al. 2023 American Heart Association Focused Update on Adult Advanced Cardiovascular Life Support: An Update to the American Heart Association Guidelines for Cardiopulmonary Resuscitation and Emergency Cardiovascular Care. Circulation. 2024;149(5):e254-e273. doi:10.1161/CIR.0000000000001194

2. Were other forms of literature such as expert consensus documents considered while reviewing the guidelines?

3. Kindly provide a more comprehensive explanation of the Data extraction and abstraction process and the statistical analysis in Methodology. Kindly refer to the following articles for reference:

Gonzalez-Del-Hoyo M, Mas-Llado C, Blaya-Peña L, et al. Type of evidence supporting ACC/AHA and ESC clinical practice guidelines for acute coronary syndrome. Clin Res Cardiol. 2024;113(4):546-560. doi:10.1007/s00392-023-02262-9

Fanaroff AC, Califf RM, Windecker S, Smith SC Jr, Lopes RD. Levels of Evidence Supporting American College of Cardiology/American Heart Association and European Society of Cardiology Guidelines, 2008-2018. JAMA. 2019 Mar 19;321(11):1069-1080. doi: 10.1001/jama.2019.1122. PMID: 30874755; PMCID: PMC6439920.

I hope this helps in furthering the quality of your research and making it more impactful.

Thank you

**Reviewer #3: **Thank you for addressing concerns. This document now is much more readable, and appears ready for publication.

7. PLOS authors have the option to publish the peer review history of their article (what does this mean?). If published, this will include your full peer review and any attached files.

Reviewer #1: No

Reviewer #2: No

Reviewer #3: **Yes: **Peter T Morley

---

## [Author Response · Author response to Decision Letter 1]

14 Jun 2024

Editorial comments:

We have revised the manuscript to provide a more comprehensive explanation of the data extraction and abstraction process, as well as the statistical analysis, in the Methodology section. We have reviewed the following articles to enhance our methodology:

Gonzalez-Del-Hoyo M, Mas-Llado C, Blaya-Peña L, et al. (2024). Type of evidence supporting ACC/AHA and ESC clinical practice guidelines for acute coronary syndrome. Clin Res Cardiol. 113(4):546-560. doi:10.1007/s00392-023-02262-9

Fanaroff AC, Califf RM, Windecker S, Smith SC Jr, Lopes RD. Levels of Evidence Supporting American College of Cardiology/American Heart Association and European Society of Cardiology Guidelines, 2008-2018. JAMA.

Reviewer #1:

Reviewer #1 did not provide specific comments.

Reviewer #2:

We have addressed the reviewer's request for incorporating updated guidelines to make the article more current. Therefore, we reviewed and compared the latest formal update from 2023. Additionally, we have provided more detailed information on how recommendations were collected and reviewed in terms of LOE and COR through data processing in the R program. Specifically, we examined changes in the Class of Recommendation (COR) and/or Level of Evidence (LOE) for all the recommendations we studied in the 2020 edition. We also reviewed the two references recommended by the reviewer (Gonzalez-Del-Hoyo et al. and Fanaroff et al.) and ensured our methodology aligns with established practices. Lastly, we included an additional limitation of the study, namely that we did not include expert consensus documents, which should be considered in future research to provide a more comprehensive evaluation.

Reviewer #3:

We appreciate the reviewer's positive feedback on the insightfulness of our submission.

---

## [Editor Report · Decision Letter 2]

8 Aug 2024

The evidence supporting AHA guidelines on adult cardiopulmonary resuscitation (CPR)

PONE-D-24-02668R2

Dear Dr. Junedahl,

We’re pleased to inform you that your manuscript has been judged scientifically suitable for publication and will be formally accepted for publication once it meets all outstanding technical requirements.

Kind regards,

Ankur Shah

Academic Editor

PLOS ONE
---

## [Editor Report · Acceptance letter]

20 Aug 2024

PONE-D-24-02668R2 

PLOS ONE

Dear Dr. Junedahl, 

I'm pleased to inform you that your manuscript has been deemed suitable for publication in PLOS ONE. Congratulations! Your manuscript is now being handed over to our production team.

Kind regards, 

on behalf of

Dr. Ankur Shah 

Academic Editor

PLOS ONE